# A Polymer for Application as a Matrix Phase in a Concept of In Situ Curable Bioresorbable Bioactive Load-Bearing Continuous Fiber Reinforced Composite Fracture Fixation Plates

**DOI:** 10.3390/molecules26051256

**Published:** 2021-02-26

**Authors:** Artem Plyusnin, Jingwei He, Cindy Elschner, Miho Nakamura, Julia Kulkova, Axel Spickenheuer, Christina Scheffler, Lippo V. J. Lassila, Niko Moritz

**Affiliations:** 1Turku Clinical Biomaterials Centre—TCBC, Department of Biomaterials Science, Faculty of Medicine, Institute of Dentistry, University of Turku, FI-20014 Turku, Finland; artem.plyusnin@utu.fi (A.P.); liplas@utu.fi (L.V.J.L.); niko.moritz@utu.fi (N.M.); 2College of Materials Science and Engineering, South China University of Technology, Guangzhou 510006, China; he.jingwei@utu.fi; 3Leibniz-Institut für Polymerforschung Dresden e. V., D-01005 Dresden, Germany; elschner@ipfdd.de (C.E.); spickenheuer@ipfdd.de (A.S.); scheffler@ipfdd.de (C.S.); 4Medicity Research Laboratory, Faculty of Medicine, University of Turku, FI-20014 Turku, Finland; mihnak@utu.fi

**Keywords:** functionalized polylactide, bioresorbable composite matrix, light curable polymer, less-rigid fracture fixation, bioresorbable FRC

## Abstract

The use of bioresorbable fracture fixation plates made of aliphatic polyesters have good potential due to good biocompatibility, reduced risk of stress-shielding, and eliminated need for plate removal. However, polyesters are ductile, and their handling properties are limited. We suggested an alternative, PLAMA (PolyLActide functionalized with diMethAcrylate), for the use as the matrix phase for the novel concept of the in situ curable bioresorbable load-bearing composite plate to reduce the limitations of conventional polyesters. The purpose was to obtain a preliminary understanding of the chemical and physical properties and the biological safety of PLAMA from the prospective of the novel concept. Modifications with different molecular masses (PLAMA-500 and PLAMA-1000) were synthesized. The efficiency of curing was assessed by the degree of convergence (DC). The mechanical properties were obtained by tensile test and thermomechanical analysis. The bioresorbability was investigated by immersion in simulated body fluid. The biocompatibility was studied in cell morphology and viability tests. PLAMA-500 showed better DC and mechanical properties, and slower bioresorbability than PLAMA-1000. Both did not prevent proliferation and normal morphological development of cells. We concluded that PLAMA-500 has potential for the use as the matrix material for bioresorbable load-bearing composite fracture fixation plates.

## 1. Introduction

Use of polymers in biomedical applications has been continuously growing in recent decades [1,2,3,4,5,6]. These materials are used to restore, replace or activate specific biological response [7,8,9,10,11,12]. Polymers can be combined with other types of biomaterials, such as ceramics, or with other polymers to form a composite with advanced performance [13,14,15,16,17,18].

Polymers include two major subgroups: biostable (i.e., non-resorbable in the body) and bioresorbable (i.e., materials that are completely dissolved and excreted from the body over time) polymers. Aliphatic polyesters, such as polylactide (PLA), polyglycolide (PGA), and their derivatives, are the most widely spread category of the bioresorbable polymers for medical applications [19,20,21,22,23,24]. Aliphatic polyesters are commercially used, for instance, in pins, screws, and plates. The use of bioresorbable fracture fixation plates made of these materials has good potential in musculoskeletal reconstructions due to their unique advantages. In general, they have good biocompatibility. In turn, gradual degradation of the plate’s material over time allows stepwise redistribution of the load-bearing function from the plate back to the bone, thus reducing the risk of stress-shielding and favoring the bone formation, which is important, for example, in the cases of elderly and pediatric patients. After the completion of its function, the plate is entirely resorbed and excreted from the body; thus, the need for the plate removal is eliminated. Moreover, in pediatric applications, bioresorbable bone fixation less hinder the natural growth of the skeleton than their biostable counterparts [25,26].

The degradation rate and mechanical properties of aliphatic polyesters are versatile to some extent and can be tailored to a particular application. This is achieved by adjusting, for example, the average molecular mass, ratio of stereoisomers (e.g., l- and d- isomers in PLA) or copolymers (e.g., lactides/glycolides) in the final polymer [27]. Despite that, the maximum achievable strength and stiffness are insufficient for many load-bearing applications. The thermoplastic nature of polyesters determines their susceptibility to creep under non-critical, but continuous load [28,29,30]. Contouring of polyester plates is possible, but usually requires additional manipulations, such as heating in a hot water bath up to the glass transition temperature (40–60 degrees) [31,32]. Moreover, the degradation of a massive bulk of polyester increases the acidity in the surrounding tissues and may lead to adverse tissue reactions [33,34,35].

In this study, an alternative for conventional thermoplastic polyesters for the use in load-bearing fracture fixation applications was suggested—functionalized thermosetting polyesters.

Functionalized thermosetting polyesters have been previously studied from the prospective of tissue engineering [36,37,38,39]. Based on the preceding research, we proposed a straightforward synthesis route of PLA functionalized with diMethAcrylate (PLAMA). At normal conditions, PLAMA represents a liquid viscous resin. From the widely used acrylic dental restoration materials, such as bisphenol-A-glycidylmethacrylate (bisGMA), the photoinitiation system was adopted [40,41,42]. This allowed curing (i.e., hardening) of PLAMA in situ using the standard dental equipment. Considering the advantages of PLAMA, we proposed a special concept of the in situ curable bioresorbable bioactive load-bearing fiber reinforced composite (FRC) fracture fixation plate. In those plates, PLAMA will be used as the matrix phase of FRC. Fibers made of bioactive glass (BG) [43,44,45,46,47,48] will be used to increase the mechanical properties and provide the bioactive properties for the whole plate to compensate for possible adverse effects caused by the products of the degradation of PLAMA [49,50,51,52]. The light curing will be used to harden the plate in situ after contouring it against the shape of a bone. The thermosetting character of PLAMA will provide more stable mechanical properties during the bone healing than in thermoplastic polyesters [28].

The present work is the first stage in the systematic investigation of PLAMA and its application in the concept of the in situ curable bioresorbable bioactive load-bearing FRC fracture fixation plate. Considering the proposed concept, in this study PLAMA was investigated from the standpoints of its chemical and physical properties and the biological safety.

### Clinical Concept

The proposed concept FRC fracture fixation plate comprises the matrix phase, the reinforcement phase, and the shell (Figure 1).

The matrix phase is represented by PLAMA with addition of a photoinitiation agent, which allows curing of the resin in situ using standard dental light-curing equipment.

The reinforcement phase is made of bioresorbable fibers, such as BG fibers. The reinforcement is formed using tailored fiber placement (TFP) technology. In this textile technique, a continuous fiber roving is placed, according to a pattern optimized for a particular application considering stress distribution [53,54,55,56]. This allows bypassing of the areas for screw holes and minimizes cutting of the reinforcement phase. TFP is widely used in aerospace and automotive industries; however, to the authors’ knowledge, has never been applied for load-bearing implantable devices. The use of TFP in an FRC plate maintains the structural integrity on all steps of the manufacturing process and reduces the need for mechanical processing of the composite.

Finally, the shell of the implant is a film made of a fast-resorbing polymer, such as oxidized cellulose-based materials. It encloses the constituents of the composite together so that the uncured implant can be handled in situ.

Prior to a surgery, the shell containing the reinforcement phase is put onto a fractured bone, contoured against the bone surface, and fixed with screws. Next, the shell is filled with the resin resulting in a pre-impregnated (“prepreg”) composite. Further, the prepreg is cured with light.

The concept is flexible. Thus, the concrete choice of materials for each of the components of the plate may vary, and is defined by the required mechanical and bioactive properties of the plate, provided that matrix-to-fibers adhesion is adequate.

## 2. Results

### 2.1. Overview of the Study

The present study comprised four stages (Figure 2). The goal was to investigate PLAMA from the standpoints of its chemical and physical properties and the biological safety.

In Stage 1, two modifications of PLAMA with different molecular masses (PLAMA-500 and PLAMA-1000) were synthesized in the form of liquid light-curable resins. A basic light curing protocol was applied to obtain hardened polymers. The efficiency of the curing was assessed by the degree of conversion (DC) of double bonds during the cross-linking reaction. In turn, DC was measured by means of Fourier-transform infrared (FTIR) spectroscopy.

In Stage 2, basic mechanical and thermomechanical properties of PLAMA were determined in a standardized tensile test under normal conditions and in the thermomechanical analysis (TMA). Elastic modulus, Poisson’s ratio, ultimate tensile strength (UTS), and the corresponding strain were determined in the tensile test. Glass transition temperature (*T_g_*) and coefficient of thermal expansion (CTE) were found in TMA.

In Stage 3, basic features of the biodegradation behavior of PLAMA were studied in vitro by immersion of PLAMA specimens in simulated body fluid (SBF). The mass changes in the PLAMA specimens, change of the pH value of SBF, and change in the flexural properties of PLAMA specimens (flexural modulus, ultimate flexural strength and strain, strength and strain at break) were analyzed.

In Stage 4, the biocompatibility of the novel polymer was analyzed using osteoblast-like cells seeded on PLAMA specimens. The change of the cells’ morphology was evaluated by fluorescence microscopy. The cells’ proliferation was assessed by measuring the metabolic activity of cells by one of the standardized Water-Soluble Tetrazolium salts -based assays, WST-8.

The materials used in each stage, including the controls, are summarized in Table 1.

### 2.2. Stage 1: Preparation and Chemical Characterization

Three groups were used in the Stage (PLAMA-500, PLAMA-1000 and bisGMA). BisGMA served as a control since the functional group, which provides the cross-linking of PLAMA, and the photoinitiation system, which starts the cross-linking reaction, had been adopted from bisGMA. The photoinitiation system consisted of camphorquinone (CQ) and 2-dimethylaminoethyl methacrylate (DMAEMA). The efficiency of the curing protocol and, thus, the cross-linking was assessed by DC of double bonds during the cross-linking reaction.

#### 2.2.1. Visual Characterization of Uncured Resins

Both PLAMA-500 and PLAMA-1000 resins represented transparent colorless viscous liquids. After blending with the photoinitiation system components, the liquids obtained yellow color. Visually, it was impossible to distinguish both PLAMA modifications and bisGMA resins from each other. The cross-linking of the resins resulted in light yellow transparent glassy materials. Whereas cross-linked PLAMA-500 and bisGMA were rigid, PLAMA-1000 was notably more flexible and less elastic when bent, resembling the behavior of a relatively hard elastomer.

#### 2.2.2. Measurement of the Degree of Conversion

Examples of the FTIR spectra used in the measurement of DC are given in Figure 3. Measured values of DC are presented in Table 2 and in Figure 4. Immediately after curing, both PLAMA-500 and PLAMA-1000 had significantly higher DC than bisGMA (128.2% and 132.0% from DC of bisGMA, correspondingly; *p* < 0.001 in both cases), whereas there was no significant difference between PLAMAs. Significant post-curing was detected in all groups between days 0 and 1 (108.1% from the value at day 0, *p* = 0.007; 106.0%, *p* = 0.001, and 126.8%, *p* = 0.002 in PLAMA-500, PLAMA-1000, and bisGMA, correspondingly). Between days 1 and 3, significant post-curing occurred in group PLAMA-500 only (103.8% from the value at day 3, *p* = 0.041). No further significant post-curing was found in all groups after day 3. On day 28, the difference between DC in groups PLAMA-500 and PLAMA-1000 was insignificant, whereas both groups had DC significantly higher than in bisGMA (111.3% from DC of bisGMA, *p* = 0.048 and 113.1% from DC of bisGMA, *p* = 0.011, correspondingly).

### 2.3. Stage 2: Physical Characterization

Four groups of specimens were used in the stage: PLAMA-500, PLAMA-1000, bisGMA, and poly (l-/dl-) lactide (PLDL). BisGMA served as a control since it is a thermoset that has a similar cross-linking mechanism as PLAMA, and since it has been successfully studied as a matrix phase of FRC implants for skeletal reconstruction [58,59]. PLDL was the second control since it has similar repeating units as PLAMA, but at the same time, it is a thermoplastic.

#### 2.3.1. Tensile Test

The groups PLAMA-500, PLAMA-1000, and bisGMA showed inelastic behavior (Figure 5). The stress-vs-strain curves of PLAMA-500 (Figure 5A) demonstrated the most ductile character, i.e., had strain softening before the break point. PLAMA-1000 had a flat curve resembling that of elastomers (Figure 5B), which was in line with subjective observations. The shape of the curves of BisGMA (Figure 5C) were similar to those of PLAMA-500, but demonstrated brittle fracture.

Contrary to expectations, the group PLDL demonstrated extremely brittle behavior (Figure 5D) with low strength. In three out of five specimens, fracture occurred before the onset of the nonlinear region of the stress-vs-strain curve; in the other two, the proportional limit was barely passed. Supposedly, the applied manufacturing method (vacuum molding by melting granules directly in a mold) was not appropriate for obtaining maximum possible mechanical properties; however, other methods were not feasible for our group at that moment.

Compared to bisGMA (Table 3, Figure 6A), the modulus of PLAMA-500 was insignificantly lower, whereas significantly higher than in PLAMA-1000 (83.9% of the bisGMA value for PLAMA-500 and 23.1%, *p* < 0.001 for PLAMA-1000). The modulus of PLDL was significantly higher than that of bisGMA (129.4% of bisGMA, *p* = 0.015).

Poisson’s ratio (Table 3, Figure 6B) of PLAMA-1000 was significantly higher than that of PLAMA-500 (93.1% of bisGMA, *p* = 0.018), bisGMA (*p* = 0.035), and PLDL (134.5% of the bisGMA value; *p* < 0.001). Differences in other pairs were insignificant. In all groups, the oscillations of the measured Poisson’s ratio value were uniformly decreasing with the growth of strain (Figure 7). In PLAMA-500 and bisGMA, the shapes of the Poisson’s ratio curves were similar, with slightly increasing average value (Figure 7A,C), while in a more elastomeric PLAMA-1000 there was an opposite tendency (Figure 7B). In PLDL, a barely seen decrease of the curve was detected (Figure 7D).

UTS was significantly different in all materials (Table 3, Figure 6C) with *p* < 0.001 in all pairs. BisGMA demonstrated the highest strength, whereas UTS of PLAMA-500 was 81.4% of bisGMA, in PLDL it was 62.2% and 16.6% in PLAMA-1000. In bisGMA and PLAMA-1000, UTS point corresponded to the break point on the engineering stress-vs-strain curve. In PLAMA-500, UTS corresponded to the highest point on the curve with some decline of that afterwards.

Ranking by strain at UTS was reversal with respect to the ranking by UTS itself (Table 3, Figure 6D). In PLAMA-500, this value was 117.7% from bisGMA, 112.8% of bisGMA in PLAMA-1000, and 36.1% of bisGMA in PLDL. Only strain value of PLDL was significantly different from the other groups (*p* < 0.001 in all pairs); between other groups, the difference was insignificant.

#### 2.3.2. Thermomechanical Analysis

The shapes of the thermograms obtained in the single cycle TMA were in line with the expectation for, correspondingly, the thermoset groups (PLAMA-500, PLAMA-1000, and bisGMA; Figure 8A–C) and the thermoplastic group (PLDL; Figure 8D). The glass transition point (*T_g_*) was clearly detected in all materials (Table 4).

In the multicyclic TMA, the resulting curves showed some progressive contraction of the specimens on each cycle (Figure 9). The numerical results averaged for the three full cycles were in line with the corresponding values measured in the single heating cycle TMA (Table 5).

### 2.4. Stage 3: SBF Immersion Test

#### 2.4.1. Visual Characterization

The specimens from the group PLAMA-500 had no apparent signs of degradation after 84 days. On the contrary, specimens from the PLAMA-1000 group had cavities within the bulk material, as well as some spots of corrosion on the surface.

#### 2.4.2. Mass Changes in Specimens

Before the immersion, the difference in average mass of specimens between the two groups was insignificant (Table 6, Figure 10A). After drying of the specimens removed from SBF, the average mass of PLAMA-1000 was significantly lower than that of PLAMA-500 (92.2% of the PLAMA-1000 mass, *p* < 0.001; Figure 10B). Mass losses within the groups were significant in both cases (97.0% of the initial value, *p* = 0.008 and 90.2%, *p* < 0.001 in PLAMA-500, and PLAMA-1000, correspondingly; Figure 10C).

#### 2.4.3. Changes in pH Value of SBF

Measurements of the pH value of SBF taken after the immersion test were compared by one sample *t* test with the pH value measured in the fresh SBF (Table 7). The temperature at which all measurements were done was slightly lower than the temperature during the test (33 ± 0.5 °C). The average pH value in the group PLAMA-1000 was significantly lower than in PLAMA-500 (95.7% of the PLAMA-500 value; *p* < 0.001; Figure 11A). Significant drop compared with the initial level of the fresh SBF was detected also in the group PLAMA-1000 (95.9% of the initial value, *p* < 0.001; Figure 11B) while no drop was observed in PLAMA-500.

#### 2.4.4. Changes in the Flexural Properties of Materials

Numerical results of the three-point bending test are summarized in Table 8. The test demonstrated different flexural behavior of the two modifications of PLAMA in both time points, which, in general, was in line with the behavior of the corresponding materials detected in the tensile tests (Figure 12A,B). Both materials exhibited inelastic response; PLAMA-500 specimens had a softening portion of stress-strain curves (Figure 12A). For dry PLAMA-1000, in the conditions of the experimental setup, a break point was not obtained in two out of five cases due to the flexibility of the material (Figure 12B). After the immersion, both materials became more brittle, and all PLAMA-1000 specimens reached a break point (Figure 12D).

At both time points, PLAMA-1000 was significantly less stiff than PLAMA-500 (13.5% and 19.5% of PLAMA-500, correspondingly) with *p* < 0.001 (Figure 13A,B). After the immersion, the stiffness within both groups significantly increased (112.8% and 162.2% from the initial value in PLAMA-500 and in PLAMA-1000, correspondingly; both *p* < 0.001; Figure 13C). The ultimate flexural strength at both time points was significantly lower in PLAMA-1000 than in PLAMA-500 (21.3% and 21.4% from PLAMA-500, correspondingly; both *p* < 0.001; Figure 13D,E). After the test, the strength increased within both groups with *p* = 0.002 for PLAMA-500 (111.6% from the initial value) and *p* = 0.049 for PLAMA-1000 (112.2% from the initial value; Figure 13F).

At both time points, PLAMA-1000 was significantly more flexible than PLAMA-500 (strain at break was 171.2% of that in PLAMA-500, *p* < 0.001 and 128.2%, *p* = 0.017, correspondingly; Figure 14A,B). After the immersion, both materials became less flexible than in their initial condition (strain at break was 82.6% of the initial value, *p* = 0.013 for PLAMA-500 and 61.9%, *p* < 0.001 for PLAMA-1000; Figure 14C).

The ultimate flexural strength and the stress at break were not equal in dry PLAMA-500. After the immersion, the material became more brittle, and both points coincided. Thus, the stress at break significantly increased (113.8% of the initial value, *p* = 0.004; Figure 14D) while the corresponding strain decreased (68.4% of the initial value, *p* = 0.009; Figure 14E).

### 2.5. Stage 4: Biocompatibility Evaluation

Four groups of specimens were prepared: PLAMA-500, PLAMA-1000, PLDL, and stainless steel (SS). PLDL was used as a control since it has similar degradation mechanism as PLAMA and, thus, may have similar effect on the surrounding tissues. SS was used as the second control as a typical bioinert implantable orthopedic material.

#### 2.5.1. Cell Morphology Test

At all time points, angular shaped cells having bundles of actin filaments (stress fibers) were found on specimens in each group (Figure 15, Figure 16 and Figure 17).

No substantial variations in the surface area were found among different groups at all time points (Table 9, Figure 18). Within the groups, only in PLAMA-500, a significant increase was detected between the time points 4 and 8 h (155.2% of the initial value, *p* = 0.015); by 24 h this rise was insignificantly compensated.

Similarly, the aspect ratio did not vary significantly among the groups (Table 10, Figure 19). Only at the time point 8 h, statistical analysis confirmed a significant difference between PLAMA-1000 and PLDL (194.1% of the PLDL value, *p* = 0.020). Within the groups of the novel polymer, some tendency of the aspect ratio to increase over time, i.e., a tendency to a more elongate shape of a cell, was observed. However, these changes were insignificant; in PLDL, there was a significant drop between 4 h and 8 h (70.8% of the initial value; *p* = 0.015), which was compensated by 24 h (147.1% of the value at 8 h, *p* = 0.017).

#### 2.5.2. Cell Viability Test

Absorbance at 450 nm measured in the cell viability test was proportional to the amount of living cells in a well. At the initial point, absorbance was significantly different only in PLAMA-1000 compared to PLDL (53.4% of the control PLDL value, *p* = 0.007). At 96 h point, absorbance in PLAMA-500 (74.2% of the PLDL value) was significantly lower than in the SS group (146.1% of PLDL, *p* = 0.022); absorbance in PLAMA-1000 was significantly lower than in PLDL (52.9% of PLDL, *p* = 0.023) and in SS (*p* < 0.001). Statistically significant growth of absorbance and, thus, of the amount of living cells was detected in all groups of materials (170.4% of the initial value, *p* = 0.029 for PLAMA-500; 133.5%, *p* = 0.020 for PLAMA-1000; 134.7%, *p* = 0.015 for PLDL; 307.8%, *p* < 0.001 for SS; Table 11, Figure 20C).

## 3. Discussion

In this study, a concept of the in situ curable bioresorbable bioactive load-bearing FRC plate for musculoskeletal reconstruction was proposed. A concept plate possesses the advantages of a less-rigid bioresorbable fixation plate, and at the same time, can be contoured to bone as a conventional metal plate. As the first milestone in the development of the concept, a candidate light-curable bioresorbable polymer for the use as the matrix phase of the composite, named PLAMA, was synthesized. In a combination of pilot experiments, its basic chemical and physical properties and the biological safety were investigated. The results of the study showed potential of PLAMA for the intended applications.

Introduction of a new treatment solution into the real medical practice requires optimal functioning of it from the standpoints of the biological safety and biomechanical efficiency, whereas the production costs remain reasonable. We believe that the selected format of a pilot overview study is highly important and useful for planning of deeper investigations of particular aspects of the proposed concept. This format limited, to some extent, the time frames and group sizes. Particularly, a limitation of this study was the small sample size in TMA, which did not allow performing statistical tests on the corresponding results. However, we consider that the obtained results gave the general preliminary understanding of the basic chemical and physical properties and the biological safety of PLAMA. In future studies, the number of specimens in each experiment should provide the possibility to perform necessary statistical tests.

Beside the major requirements of the biological safety and efficiency, in the choice of the composition of a polymer, we aimed at the simplicity, reproducibility, and minimization of the costs of the synthesis process. Hence, a straightforward route applying commercially available PLA diols was employed to prepare the basic oligomer mixtures and, consequently, light-curable resins. Thermosetting polyester oligomers have been previously obtained by functionalization of polyester polyols with unsaturated methacrylate [36,37,39,60], acrylate [38], or fumarate [61] groups. In the present study, the methacrylate group, used in approved dental and orthopedic materials, such as bisGMA or polymethyl methacrylate (PMMA), was chosen for the functionalization of PLA. The efficiency of the curing and, thus, the formation of cross-links between functional groups, measured by DC, in turn, largely depends on the composition of the photoinitiation system and the selected light curing protocol and strongly affects the physical properties of the cured polymers [62,63]. For PLAMA, a combination of CQ and DMAEMA was selected as a photoinitiation system. That combination is also widely used in dental polymers, such as bisGMA, and provides effective cross-linking reaction [40,41,42]. Moreover, it has been shown that the effectiveness of the cross-linking in FRC based on dimethacrylic matrix with CQ/DMAEMA photoinitiation system is not significantly affected by the presence of bone tissues and blood [64], which is important for the suggested plate concept. Additionally, a wide range of already existing and easily available light-curing equipment can be used to cure a plate in situ. It is worthy of note that the DC obtained in the present study for PLAMA was higher than that in the prototype dimethacrylic resin bisGMA. In turn, among the values of DC for bisGMA found in the literature [40,65,66,67], the highest DC (80.6% on day 7 after curing) was obtained exactly with CQ/DMAEMA photoinitiation system.

Preparation of two modifications of PLAMA from the precursor diols with different molecular masses (500 and 1000 Da) allowed observation of the change in the properties as a function of the molecular mass of the polymer. Analogously to the observation found in the literature [39], all experiments conducted in the study showed significant difference in most of the properties between PLAMA-500 and PLAMA-1000. Supposedly, by mixing of the diols in different proportions, the average molecular mass of the obtained PLAMA can be gradually altered in between the two initial masses of the applied diols. This can be used to smoothly adjust the properties of PLAMA, and, therefore, the mechanical and biodegradation properties of the whole plate can be tailored according to a particular clinical application and the corresponding demands of a surgeon. For example, the mechanical properties of PLAMA-500 were significantly higher than that of PLAMA-1000. On the other hand, the degradation rate, which can be indirectly assessed by the mass losses in polymer specimens and the effect on the pH value of SBF after the immersion test, was significantly higher in PLAMA-1000. Both observations might be attributed to the opposite effects of one phenomenon: the increase in the polymer’s crosslinks density and the corresponding decrease in the chain length leads to higher stiffness and strength [68], while a lower crosslinks density and a higher chain length results in faster degradation [68,69]. It should be noted that only the starting degradation of the two PLAMA modifications was examined in the framework of the present study. Further experiments are needed to investigate the period of complete degradation.

In terms of the mechanical performance of the novel polymer, the highest stiffness and strength values shown by PLAMA-500 were close to those of bisGMA, which, in turn, has been studied in vivo as a matrix phase in load-bearing composite implants reinforced with conventional E-glass fibers [58,59]. Thus, one may expect sufficient mechanical performance from PLAMA-based FRCs for load-bearing applications. Provided that the experimental data for the reinforcing fibers of FRC are available, the data achieved in the mechanical testing of PLAMA may serve for the preliminary simulation of the mechanical behavior of novel FRC implants by means of finite element modelling (FEM).

From the standpoint of the biological safety, two main aspects related to the chemical structure of PLAMA should be taken into account. First, the main mechanism of the degradation of PLAMA in the body is provided by the cleavage of the bonds between repeating LA units and release of those into surrounding fluids and tissues similarly to the degradation of conventional PLA [68]. This leads to a drop of the pH value of the fluids [70,71,72]. Possible reactions of the organism to the increased acidity are well documented and may lead to the formation of osteolytic zones in the damaged bone [33,34,35]. Nevertheless, the in vitro biocompatibility evaluation conducted within the present study showed the significant proliferation of osteoblast-like MG-63 cells in the presence of PLAMA. This cell line is widely used in biocompatibility studies of materials for orthopedic applications [73,74,75]. In the present study, angular shape and formation of stress fibers in the cells were visually detected, which is important for confirmation of normal morphogenesis, adhesion, and migration of cells. The aspect ratio and the cells’ surface area in all groups, including the control stainless steel and PLDL groups did not differ significantly. Thus, one may conclude that PLAMA did not affect the morphology of cells differently from the conventional materials. In general, the present study confirmed that normal functioning and development of cells in the presence of PLAMA was possible in vitro. Together with similar cytotoxicity experiments reported for analogous PLA-based cross-linkable resins studied from the prospective of tissue engineering [36,37,38], the biocompatibility of PLAMA looks promising. Nevertheless, more detailed studies, such as analysis of the degradation products and the correlation of their release with the results of in vitro experiments, are desirable.

Nevertheless, in the next step of the research of the proposed concept, BG fibers will be used [43,44,45,46,47,48] as the reinforcement phase of a bioresorbable bioactive FRC. Being in contact with body fluids, BG is capable of increasing the fluids’ pH value due to the release of alkali ions through a cascade of chemical reactions [76,77]. This would buffer the expected local drop in pH value caused by acidic degrading polymers such as PLAMA.

In addition to the pH compensatory effect, the use of BG as a bioactive agent can stimulate bone growth [49,50,51,52] and provide antimicrobial effect [78,79]. Hence, BG and other bioactive ceramics have been logically scrutinized as a bioactive filler within a polyester matrix in composite implants and scaffolds in multiple studies [76,80,81,82,83,84,85,86]. Thus, in the suggested novel plate concept, the reinforcement phase made of BG fibers will provide both load-bearing and bioactive functions. By varying the composition of BG, amount, and distribution of fibers, the degradation rate and the bioactivity of the reinforcement phase can be adjusted in accordance with the clinical requirements providing the favorable biological response during all steps of the healing process.

The second aspect of the biological safety of PLAMA is the effect of the functional methacrylate groups presented on the ends of the molecular chains. Structural acrylate-based biomaterials have been in clinical practice for decades since the introduction of PMMA in the middle of the 20th century. Therefore, the adverse effects of acrylates are well studied in dentistry and orthopedics and include skin allergic reactions [87,88,89,90], cytotoxicity of acrylate free radicals [91], inflammatory reaction, and osteolysis caused by wearing particles of a bulk acrylate [92,93,94]. However, acrylates are still in demand, and studies on novel acrylate-based biomaterials are still being reported [95,96,97]. Acrylic materials used for decades in orthopedic and dental restoration, such as PMMA and bisGMA, have even been used in such sensitive applications as cranial implants [98,99,100].

## 4. Materials and Methods

### 4.1. Stage 1: Preparation and Chemical Characterization

#### 4.1.1. Synthesis of Oligomer Mixtures 

The synthesis route for PLAMA is schematically depicted in Figure 21. For PLAMA-500, the mixture was prepared based on 50 g of commercially available PLA diol with the molecular mass 500 (PLA205B, Shenzhen eSUN Industrial Co., Ltd., Shenzhen, China), whereas the mixture for PLAMA-1000 was based on 100 g of PLA diol with the molecular mass 1000 (PLA210B, Shenzhen eSUN Industrial Co., Ltd., Shenzhen, China). Unlike the modification with a high molecular mass (2000), these diols could maintain relatively low viscosity at normal conditions. Then, 31.04 g of 2-isocyanatoethyl methacrylate and two droplets of di-*n*-butyltin dilaurate (both—Tokyo Chemical Industry Co., Ltd., Tokyo, Japan) were added to either mixture. Next, the mixtures were stirred to obtain a homogenous structure. Subsequently, 50 mL of acetone were added into either mixture to reduce the viscosity, and the temperature of the mixtures was raised up to 45 °C. The reaction was continued until the infrared absorbance peak of the -NCO group (2270 cm^−1^) disappeared in FTIR spectra of the samples taken from either mixture. Vector 33 FTIR instrument (Bruker Corp., Billerica, MA, USA) was used. After removing the acetone by distillation in vacuum, viscous colorless liquids with a yield of 100% were obtained. The molecular structures of both PLAMA modifications were controlled by FTIR spectroscopy and proton nuclear magnetic resonance (^1^H-NMR) spectroscopy (AVANCE AV 400 MHz, Bruker Corporation, Billerica, MA, USA; Table 12). ^1^H-NMR was done at 400 MHz, using deuterated chloroform (CDCl_3_) as a solvent.

#### 4.1.2. Preparation of Light-Curable Resins

Light-curable resins were prepared by blending either of PLAMA-500 and PLAMA-1000 oligomer mixtures with ethylene glycol dimethacrylate in a mass ratio of 80/20. Then, 0.7 wt.% of CQ (Sigma-Aldrich Corp., St. Louis, MO, USA) and 0.7 wt.% of DMAEMA (Fluka Chemie GmbH, Buchs, Switzerland) were added as a photoinitiation system. The reagents were heated to 45 °C and stirred for at least 6 h to obtain a homogenous blend.

#### 4.1.3. Light Curing Protocol

The protocol for the light curing of liquid resins was based on our previous work [57]. Liquid resin was poured in a corresponding mold, then degasified by placing the mold for one hour at 37 °C–39 °C in an oven and, subsequently, in a vacuum chamber at 25 °C and 10 mbar for 30 min. Then, the specimens were pre-cured with a hand light curing device (Elipar S10, 3M ESPE, Seefeld, Germany) for two minutes. Next, the pre-cured specimens were taken out of the mold and placed in a light oven (Targis Power TP1, Ivoclar AG, Schaan, Liechtenstein) at the standard program #1 (25 min in total, maximum temperature 104 °C) and in a vacuum light oven (Visio Beta Vario, 3M ESPE, Seefeld, Germany) at the standard program #00 (15 min, 1 mbar).

#### 4.1.4. Measurement of the Degree of Conversion

A bottomless mold with a cylindrical cutout with the diameter 6 mm and the height 3 mm was made from lab putty and placed on the attenuated total reflection detector of a FTIR instrument (Frontier, PerkinElmer, Inc., Waltham, MA, USA). Liquid resin was poured into the cutout of the mold, and the FTIR spectrum was measured. Next, the resin was pre-cured directly on the FTIR detector with a hand light curing device, and further removed and exposed to the remaining stages of the curing protocol. Spectrum was measured immediately after the full curing cycle and at the time points: one, three, seven, and 28 days after curing.

DC was calculated from the ratios of the aliphatic peak (C=C) at 1636 cm^−1^ and the carbonyl peak (C=O) at 1720 cm^−1^ presented in the methacrylate group, in the cured and uncured states, as described in previous studies [65,66,67]:(1)$$DC\left(t\right)=\left[\frac{1-{\left(A_{C=C}/A_{C=O}\right)}_{t}}{{\left(A_{C=C}/A_{C=O}\right)}_{0}}\right]\times 100\%,$$
where DC(t) is the degree of conversion as a function of time; $A_{C=C}$ and $A_{C=O}$ are the aliphatic and carbonyl peaks of the spectrum.

### 4.2. Stage 2: Physical Characterization

#### 4.2.1. Preparation of Specimens for Tensile Test

Type 1BA specimens were prepared according to the standard ISO 527-2. The aimed nominal thickness was 3 mm.

The specimens from the groups PLAMA-500, PLAMA-1000, and bisGMA were vacuum molded from the prepared liquid resins in a polytetrafluoroethylene (PTFE) mold and light cured as described for Stage 1. The PLDL specimens were vacuum molded in a silicone mold. For that, pellets of analytical grade PLDL with the ratio 70/30% (PURASORB PLDL 7028, Corbion N.V., Amsterdam, The Netherlands), kindly provided by the manufacturer, were put in the mold; subsequently, the mold was placed in a vacuum oven at 200 °C and 10 mbar until full melting of granules. The molded specimens from all groups were further milled to the final geometrical dimensions with a computer programmed milling machine. The flat surfaces of the specimens were polished with grinding paper #2000 without water irrigation.

Prior to testing, all specimens were dried in a vacuum chamber at room temperature (RT), and pressure 10 mBar for 72 h. Right after the removal from the vacuum, a high-elongation gage for measuring longitudinal strain (KFEL-5-120-C1L3M3R, Kyowa Electronic Instruments Co., Ltd., Chofu, Japan) was attached in the middle of each specimen on one of the flat surfaces. Another one, for transverse strain (KFGS-02-120-C1-11L3M2R, Kyowa Electronic Instruments Co., Ltd., Chofu, Japan), was attached in the middle of the opposite surface using special glue (CC-33A, Kyowa Electronic Instruments Co., Ltd., Chofu, Japan). Before gluing, the specimens were degreased with ethanol. After gluing, the assembled specimens were stored in a desiccator for at least 1 h prior to testing, as recommended by Kyowa.

#### 4.2.2. Experimental Setup for Tensile Test

The test was conducted following the standards ISO 527-1 and ISO 527-2. Correspondingly, at least five specimens were to be tested in each group to provide robust statistics. Specimens were clamped vertically in a universal testing machine LR30K Plus (Lloyd Instruments Ltd., Bognor Regis, UK), equipped with a load cell XLC-2500-A1 (Lloyd Instruments Ltd., Bognor Regis, UK; maximum load 2500 N). The specimens were subjected to tension with a pre-loading force 1 N and a loading rate of 1 mm/min. Sensor interface PCD-300A (Kyowa Electronic Instruments Co., Ltd., Chofu, Japan) was applied to record the data from strain gages.

The strain-vs.-time curve received from PCD-300A was combined with the load-vs.-time curve recorded with LR30K Plus to obtain the final engineering stress-vs.-strain curve. From the recorded data, the required material parameters were calculated as described in the corresponding standards.

#### 4.2.3. Preparation of Specimens for Thermomechanical Analysis

Four groups of specimens were used (PLAMA-500, PLAMA-1000, PLDL, and bisGMA). One of the intact specimens made for the tensile test was taken from each group and cut into parallelepiped pieces with the dimensions 5 × 5 × 3 mm. Prior to the analysis, the specimens were dried in vacuum at 50 °C and 50 mbar for four hours and then stored in a desiccator until the measurement.

#### 4.2.4. Experimental Setup for Thermomechanical Analysis

Thermomechanical analyzer Q400 (TA Instruments, New Castle, DE, USA) was used in TMA. During the test, a specimen was placed in a sample chamber and subjected to a constant compressive force of 0.03 N from the top via a probe. The specimen was cooled down to −25 °C and then heated up to 100 °C (for the PLDL group) or 250 °C (for all other groups) at a rate of 2 °C/min. The change in the thickness of the specimen was measured with an inductive measuring system. TMA was performed in two steps. In the first step (single cycle TMA), a specimen was heated and measured once. In the second step (multicyclic TMA), a specimen was heated and cooled down three times to observe the stability of the material’s thermomechanical properties.

For each tested specimen, a thermogram (the specimen’s thickness change versus the temperature increase) was recorded. The linear portions of the obtained curve were approximated with straight lines; the intersection points of those corresponded to transition points. Mean linear CTE were calculated as the changes of the initial thickness of a specimen corresponding to a temperature increase from 18 °C to 22 °C and from 35 °C to 45 °C.

In the multicyclic TMA, the thermomechanical properties were calculated analogously to the single cycle TMA on each of the three heating portions of the curves.

### 4.3. Stage 3: SBF Immersion Test

#### 4.3.1. Preparation of Specimens

Two groups of specimens were used (PLAMA-500 and PLAMA-1000). The specimens were vacuum molded in a PTFE mold as described for Stage 1. The specimens had a parallelepiped shape with the dimensions 20 × 2 × 2 mm.

#### 4.3.2. Preparation of SBF

SBF was prepared according to Kokubo and Takadama [101] and sterilized with a sterile filter with a pore size 0.2 µm (VWR International, Radnor, PA, USA) prior to the experiments.

#### 4.3.3. Experimental Setup and Data Acquisition

The polymer specimens were immersed in SBF in sterile plastic tubes for 84 days at 37 °C. The method suggested by Kokubo and Takadama [101] for BG was adapted for the experiment. Accordingly, the necessary volume of liquid was estimated considering the area of the outer surface of a specimen as 21 mL per specimen. Tubes with pure SBF served as baseline. The tubes were placed in a shaking water bath (OLS 200, Grant Instruments, Shepreth, UK); the shaking rate was 50 cycles per minute. Control polymer specimens were placed in a desiccator at RT and continuously kept there up to the weighing.

To determine mass changes, all specimens were weighed prior to and after the immersion in SBF. After removal from SBF, the specimens were rinsed with distilled water and blown with compressed air. Next, after drying for 96 h at 37 °C in a hot air oven and drying for 24 h at RT and 10 mbar in a vacuum oven, the specimens were weighed.

Measurement of pH value was done immediately after the removal of the specimens while the temperature of the liquid remained 37 °C. A laboratory pH meter was employed (MeterLab PHM 220; Radiometer Copenhagen, Copenhagen, Denmark).

A three-point bending test was conducted to determine flexural properties following the standard ASTM 790-03. A universal testing machine (LR30K Plus; Lloyd Instruments Ltd., Bognor Regis, UK) was used. The load-deflection curves were recorded; flexural modulus, ultimate flexural strength, and strain at ultimate flexural strength, stress, and strain at break were derived from these data.

### 4.4. Stage 4: Biocompatibility Evaluation

#### 4.4.1. Preparation of Specimens

For the groups bisGMA, PLAMA-500, and PLAMA-1000, plates with the dimensions 65 × 4 × 1.7 mm were vacuum molded from liquid resins and light-cured as described for Stage 1. For the group PLDL, a plate with the same dimensions was vacuum molded in a PTFE mold as described for Stage 2. The detachable bottoms of the molds were polished with grinding paper #4000 so that one of the surfaces of the resulting plates had a smooth working surface. After molding, the acquired polymeric plates were cut to obtain parallelepiped specimens having a smooth working surface and the dimensions 4 × 4 × 1.7 mm. Thus, specimens fit a standard 96-well plate. The working surface area of a specimen was 16 mm^2^. All polymeric specimens were sterilized by gamma-irradiation.

For the group SS, specimens with the dimensions 4 × 4 × 1.7 mm were cut directly from a commercially available fracture fixation plate (Synthes AG, Solothurn, Switzerland) using a milling machine. Working surfaces of the metallic specimens were polished with grinding paper #4000 with water irrigation. The specimens were later washed with ethanol and sterilized by autoclaving. The same type of specimens was used in both experiments of Stage 4.

#### 4.4.2. Preparation of Cells

Human osteosarcoma cells from the line MG-63 were used. Prior to the tests, the cells maintained in a Petri dish in Dulbecco’s modified eagle medium (Gibco 21969035, Fisher Scientific, Waltham, MA, USA), supplemented with 10% Fetal Bovine Serum (FBS), 100 U/mL Penicillin-Streptomycin (Gibco 15140122, Fisher Scientific, Waltham, MA, USA) and 2 mM L-glutamine (Gibco 25030024, Fisher Scientific, Waltham, MA, USA) in a humidified atmosphere of 5% CO2 at 37 °C. The medium was changed every 3–4 days. For the subsequent experiments, the cells were detached from the surfaces of the Petri dish after reaching 70% confluence by treatment with 0.05% trypsin-EDTA (Gibco 25300024, Fisher Scientific, Waltham, MA, USA). Next, material specimens were washed with cell culture medium and placed in 96-well plates, followed with the seeding of the prepared cells on the top surfaces of the specimens at a density of 5000 cells per specimen. The cell preparation procedures were the same for both tests of Stage 4.

#### 4.4.3. Cell Morphology Test

The test included three time points (4, 8, and 24 h). At each time point, three specimens from each group were used. After seeding, the cells were cultured in cell culture medium in a humidified atmosphere of 5% CO2 at 37 °C for a corresponding time length.

After the removal of the medium at the corresponding time point, the specimens were twice washed with PBS (Gibco 10010015, Fisher Scientific, Waltham, MA, USA). Thereafter, the cells were fixed on the specimens using 4% paraformaldehyde in PBS for 20 min. Subsequently, the cells were permeabilized with 1% Triton X-100 (Sigma-Aldrich, St. Louis, MO, USA) in PBS for 5 min. Then, the cells were stained with rhodamine phalloidin (Invitrogen R415, Fisher Scientific, St. Louis, MO, USA) for 30 min and with Hoechst 33258 (Sigma-Aldrich) for 10 min. After stepwise application of each reagent, the specimens were washed with PBS. Each stained specimen was placed on a microscope slide into a drop of 70% solution of glycerol in PBS for subsequent observation with a microscope. All procedures were done at RT.

Stained cells were observed using a fluorescence microscope (Leica DMRB, Leica Camera AG, Wetzlar, Germany). Images were acquired with a digital camera (Olympus DP72, Olympus Corporation, Shinjuku City, Japan) and a special software package (CellSens Entry 1.5, Olympus Corporation, Shinjuku City, Japan).

The cell morphology was examined visually and quantitatively using the acquired digital images (N = 10 cells per group). Quantitative analysis included calculation of the surface area of a cell and the aspect ratio between the longitudinal and transverse dimensions of a cell. The longest distance between the edges of a cell was considered as the longitudinal dimension. The transverse dimension was measured perpendicularly to the longitudinal one between the outermost edges (Figure 22).

#### 4.4.4. Cell Viability Test

The test included two time points (24 and 96 h). For each time point, six specimens from each group were used. Six wells per well plate were seeded with cells without placing specimens and served as a baseline. After seeding, the cells were cultured in cell culture medium in a humidified atmosphere of 5% CO_2_ at 37 °C for a corresponding period of time.

Cell counting kit WST-8 (Dojindo CK04, Dojindo Laboratories, Kumamoto, Japan) was used. At the corresponding time points, WST-8 was added; thereafter, the cells were incubated for 1 h in a humidified atmosphere of 5% CO_2_ at 37 °C. After the incubation, 1% sodium dodecyl sulfate was added. Subsequently, the resulting solution was moved to a clean well plate. The absorbance at 450 nm was measured using a microplate reader Wallac 1420 Victor2 (PerkinElmer, Inc., Waltham, MA, USA).

### 4.5. Statistical Analysis of the Numerical Results

A commercial software package (SPSS Statistics 25, IBM Corporation, Armonk, NY, USA) was used for the statistical analysis, when appropriate. The normality of the distribution of the acquired data sets was evaluated with the Kolmogorov–Smirnov test. If all data sets in a statistical analysis were normally distributed, independent data sets were compared using either the independent-samples *t* test (for two data samples) or one-way ANOVA with Tukey post hoc test (for three or more data samples). If normality of the data distribution was not confirmed for at least one data set, the Mann–Whitney U test or Kruskal–Wallis test with pairwise comparison were applied, correspondingly. For the paired data sets, paired-samples *t* test (all differences between paired data sets were normally distributed) or Wilcoxon signed-ranks test (one or more differences between data sets were not normally distributed) were employed. The significance level *p* was 0.05 in all analyses. If all data sets in an analysis are normally distributed, the data are reported as the mean value and the standard deviation; otherwise, the median, 25th, and 75th percentiles are presented.

## 5. Conclusions

The first stage in the development of the proposed concept of the in situ curable bioresorbable bioactive load-bearing FRC fracture fixation plate has been done. The present study combined pilot experiments to obtain a general preliminary understanding on the chemical and physical properties and the biological safety of PLAMA. We concluded that PLAMA-500 has potential for the use as the matrix material in the concept FRC plates.

Further studies will include: a more detailed investigation of mechanical and biological properties of PLAMA in combination with BG fibers within FRC structures; development of a suitable fiber sizing for BG fibers; development of the design of FRC plate prototypes; computer simulation, real mechanical testing and in vivo investigation of the biological behavior of the FRC plates.

## 6. Patents

The concept of the in situ curable load-bearing fiber reinforced composite fracture fixation plate has been patented: Kulkova Y. Bone implant 2020 (European patent No. EP3370792B1).

## Figures and Tables

**Figure 1 molecules-26-01256-f001:**
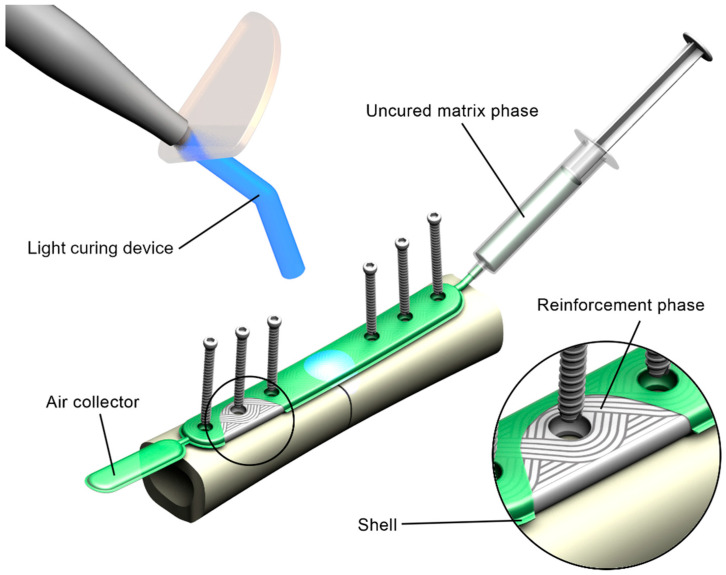
Rendering of the concept of the in situ fillable and light-curable plate used for fixation of a fracture of a long bone.

**Figure 2 molecules-26-01256-f002:**
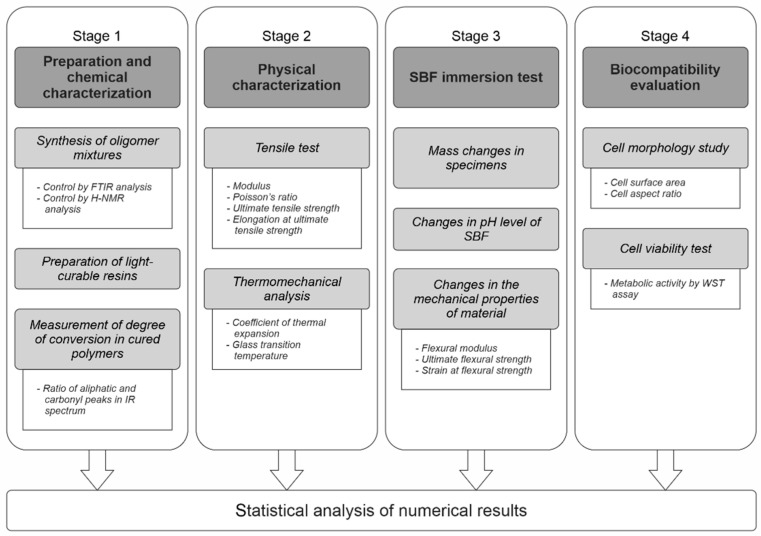
The overall workflow of the study.

**Figure 3 molecules-26-01256-f003:**
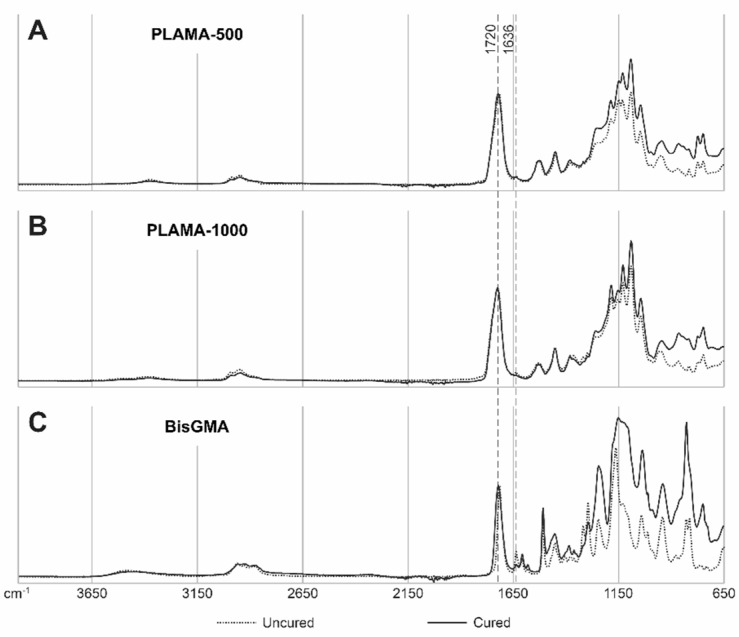
Examples of IR spectra for uncured and cured (on day 28) materials: PLAMA-500 (**A**), PLAMA-1000 (**B**), bisGMA (**C**). The curves are normalized at wavenumbers 4000 and 1720 cm^−1^.

**Figure 4 molecules-26-01256-f004:**
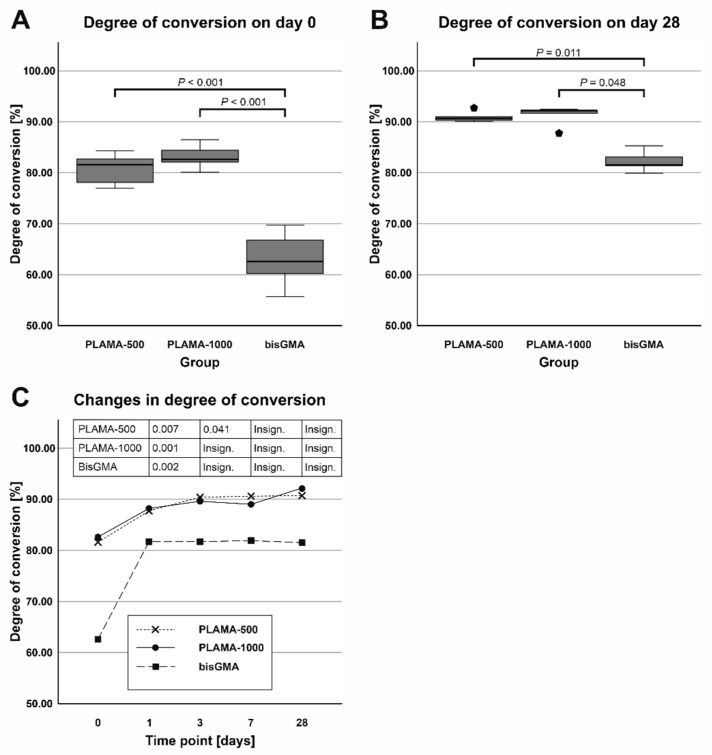
Comparison of the average degree of convergence (DC) in all groups measured on day 0 (**A**) and day 28 (**B**); changes in DC (post curing) in the specimens within 28 days after curing (**C**).

**Figure 5 molecules-26-01256-f005:**
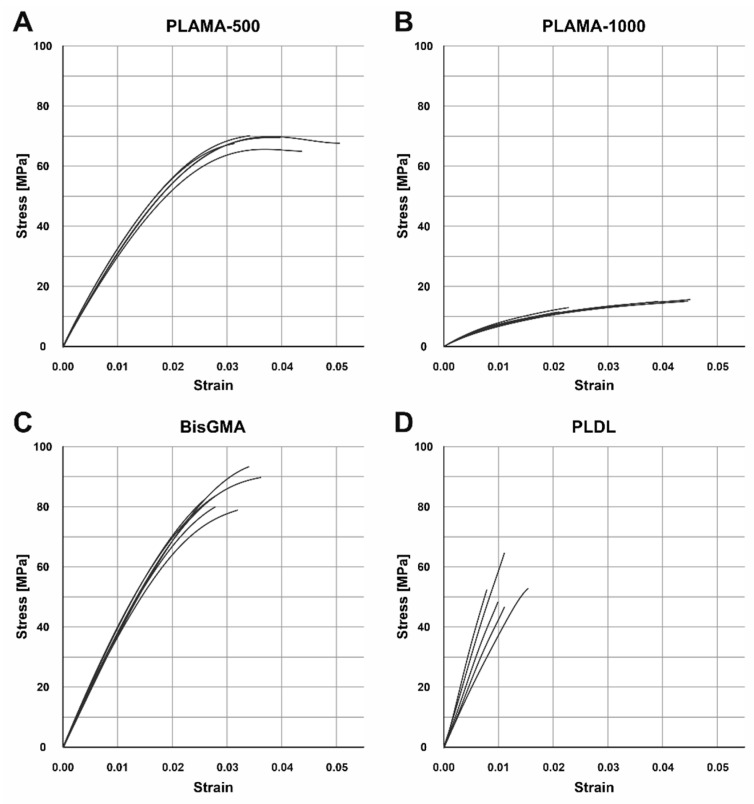
Engineering stress-strain curves of the tensile test in groups: PLAMA-500 (**A**), PLAMA-1000 (**B**), bisGMA (**C**), PLDL (**D**).

**Figure 6 molecules-26-01256-f006:**
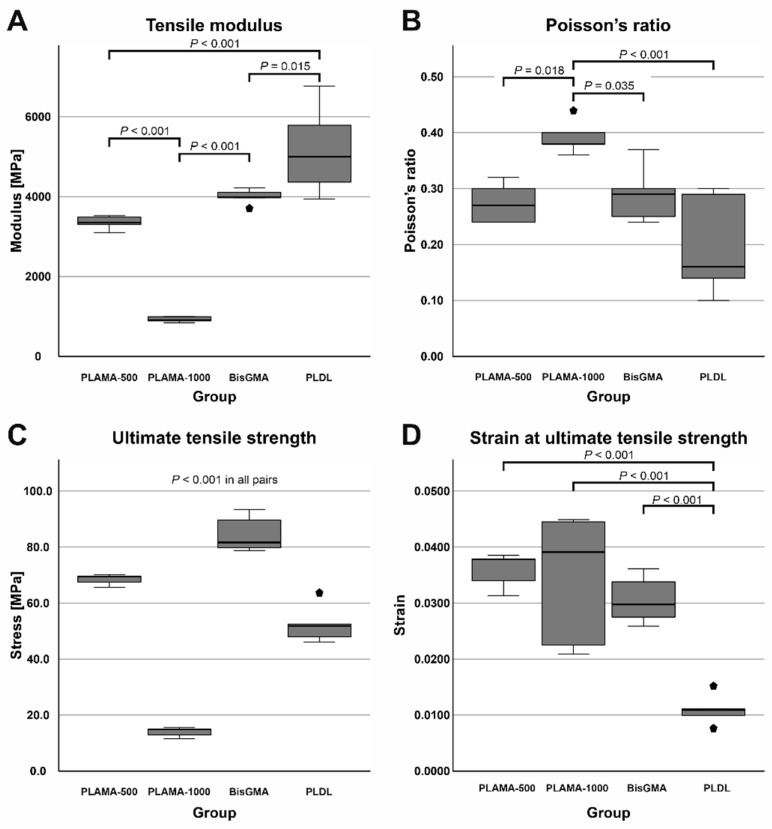
Comparison of the basic mechanical properties of the materials tested in the study: tensile modulus (**A**), Poisson’s ratio (**B**), ultimate tensile strength (UTS) (**C**), strain at UTS (**D**).

**Figure 7 molecules-26-01256-f007:**
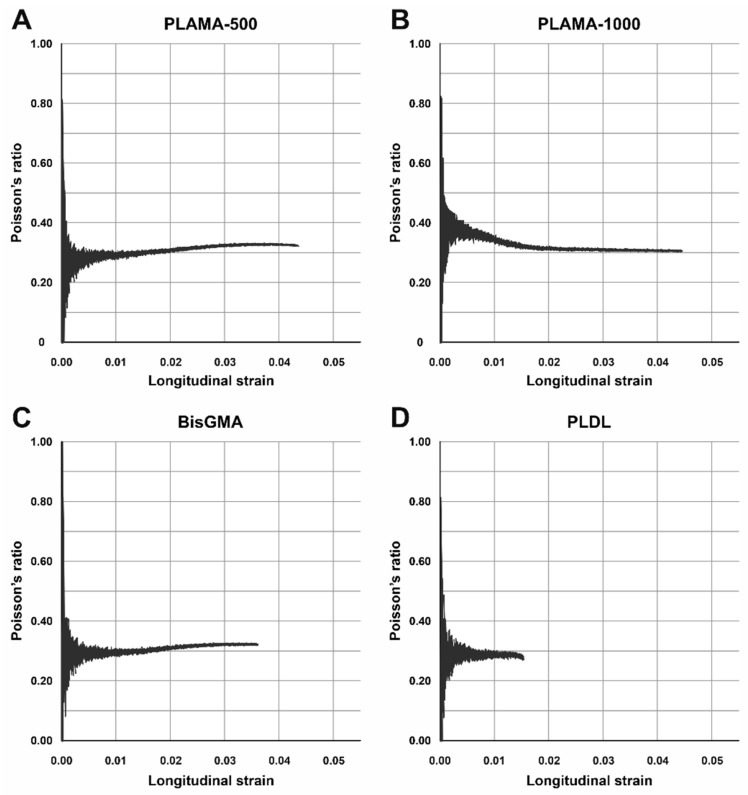
Examples of the change of Poisson’s ratio vs. longitudinal strain of the specimens in groups: PLAMA-500 (**A**), PLAMA-1000 (**B**), bisGMA (**C**), PLDL (**D**), −.

**Figure 8 molecules-26-01256-f008:**
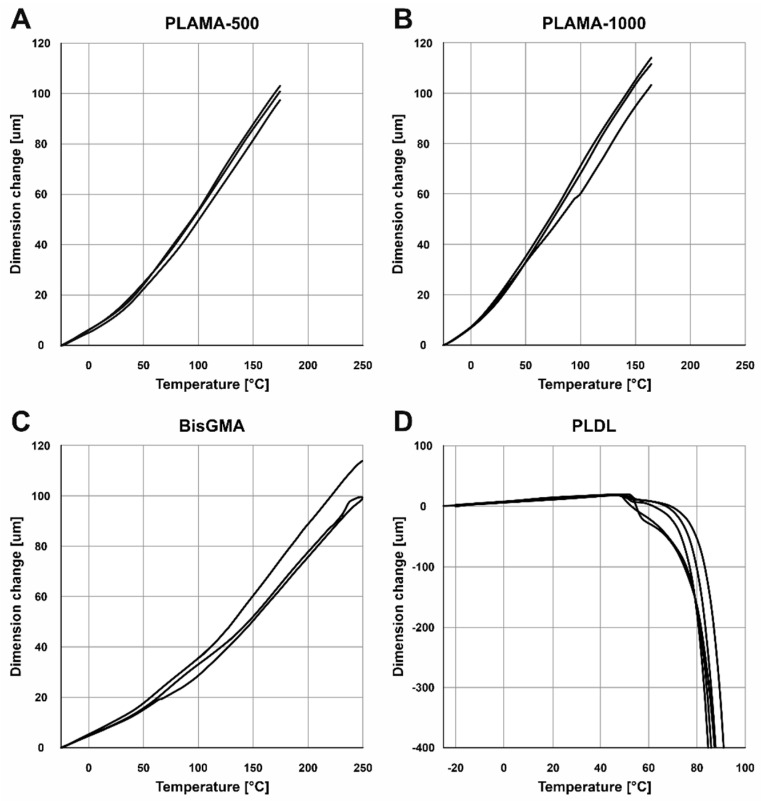
TMA thermograms for the single heating cycle in groups: PLAMA-500 (**A**), PLAMA-1000 (**B**), bisGMA (**C**), and PLDL (**D**).

**Figure 9 molecules-26-01256-f009:**
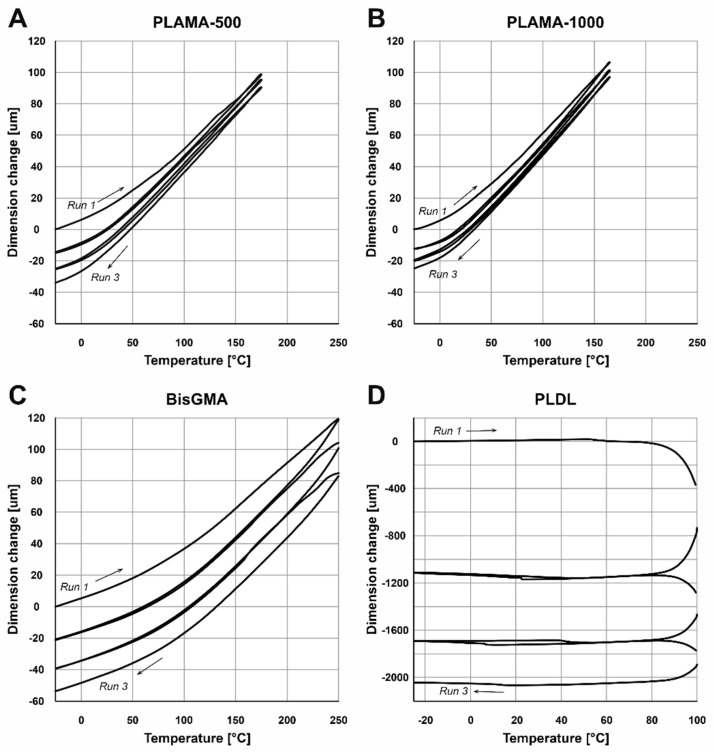
TMA thermograms for the three consecutive heating and cooling cycles in groups: PLAMA-500 (**A**), PLAMA-1000 (**B**), bisGMA (**C**), and PLDL (**D**).

**Figure 10 molecules-26-01256-f010:**
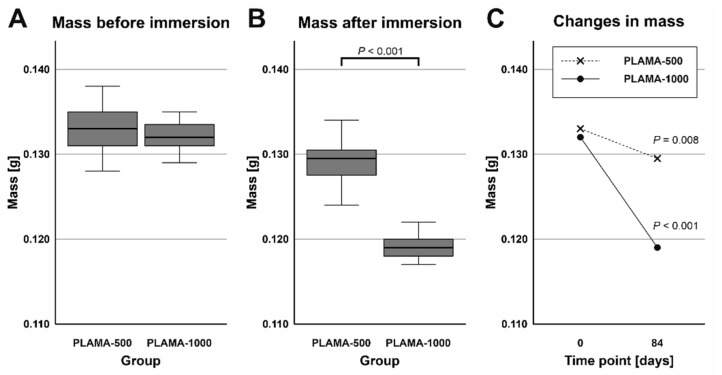
Comparison of the masses of specimens before and after the simulated body fluid (SBF) immersion test in groups PLAMA-500 (**A**) and PLAMA-1000 (**B**); comparison of changes in the masses of specimens (**C**).

**Figure 11 molecules-26-01256-f011:**
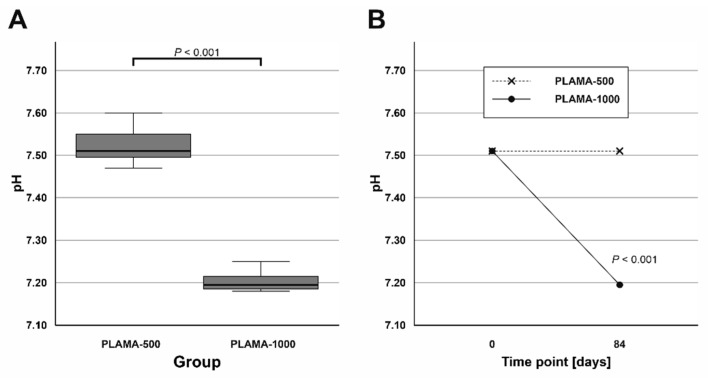
Comparison of the measured pH value in SBF in PLAMA-500 and PLAMA-1000 after the immersion test (**A**); changes in the pH level of the fluid in both groups after the test (**B**).

**Figure 12 molecules-26-01256-f012:**
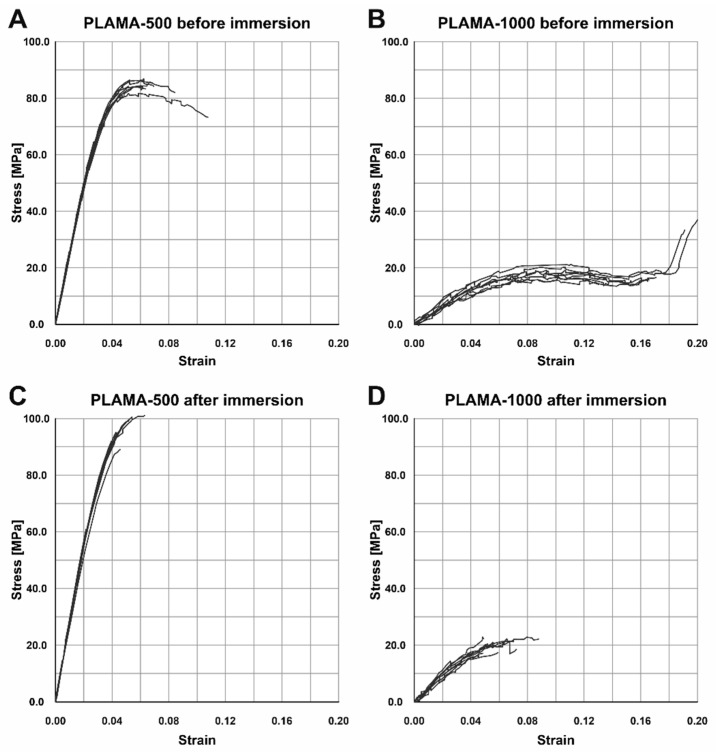
Stress-strain curves of the three-point bending test of specimens before the SBF immersion test in PLAMA-500 (**A**) and PLAMA-1000 (**B**), and after the immersion test in PLAMA-500 (**C**) and PLAMA-1000 (**D**).

**Figure 13 molecules-26-01256-f013:**
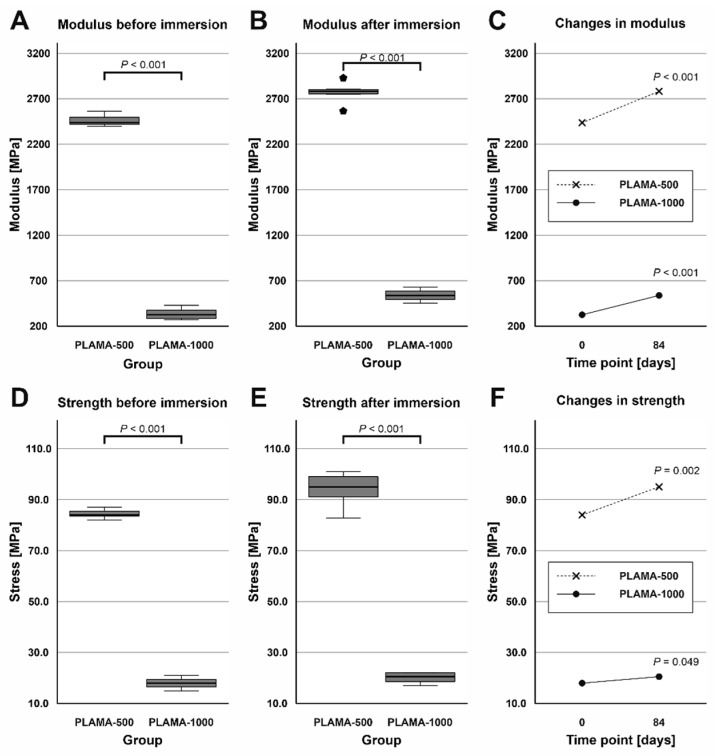
Comparison of the flexural modulus of PLAMA-500 and PLAMA-1000 before and after the SBF immersion test (**A**,**B**); changes in the modulus in both groups (**C**); comparison of the ultimate flexural strength of PLAMA-500 and PLAMA-1000 (**D**,**E**); changes in the strength in both groups (**F**).

**Figure 14 molecules-26-01256-f014:**
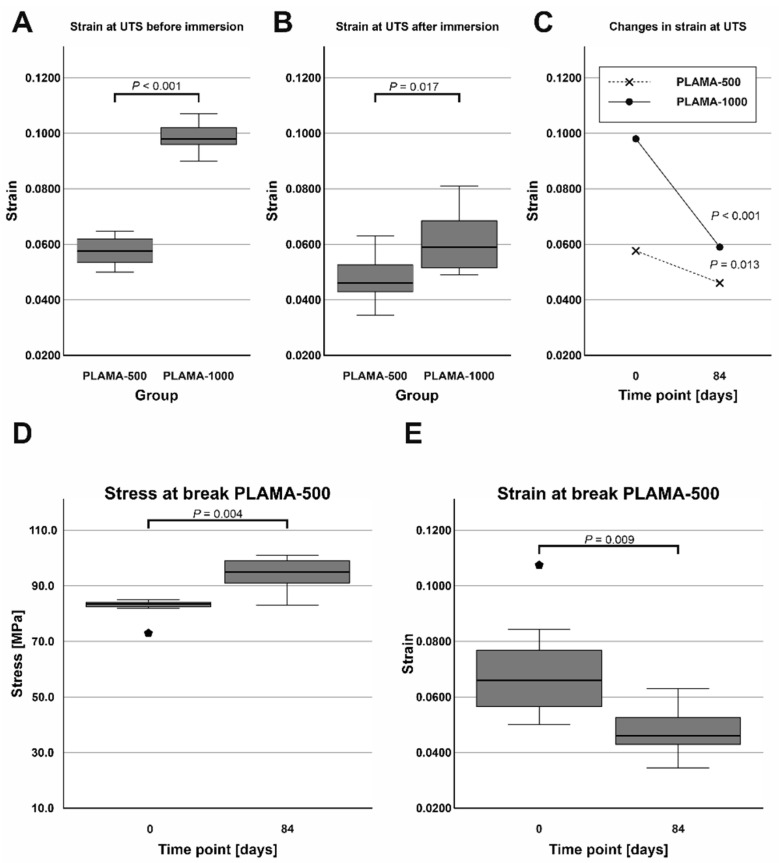
Comparison of the strain at ultimate flexural strength of PLAMA-500 and PLAMA-1000 before and after the SBF immersion test (**A**,**B**); changes in the strain in both groups (**C**); changes in the stress at break (**D**) and the corresponding strain (**E**) in PLAMA-500.

**Figure 15 molecules-26-01256-f015:**
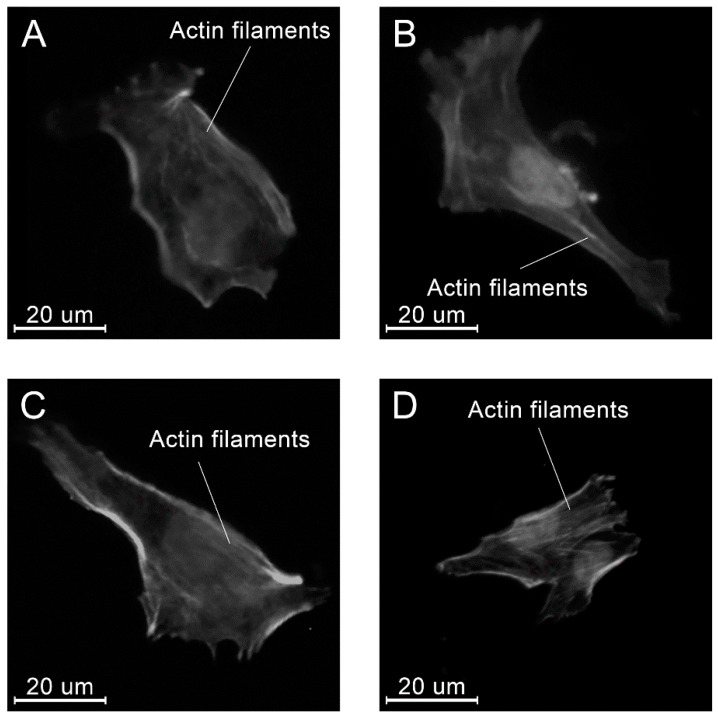
Examples of the morphology of the MG-63 cells, 4 h after seeding on the specimens of different materials: PLAMA-500 (**A**); PLAMA-1000 (**B**); PLDL (**C**); SS (**D**).

**Figure 16 molecules-26-01256-f016:**
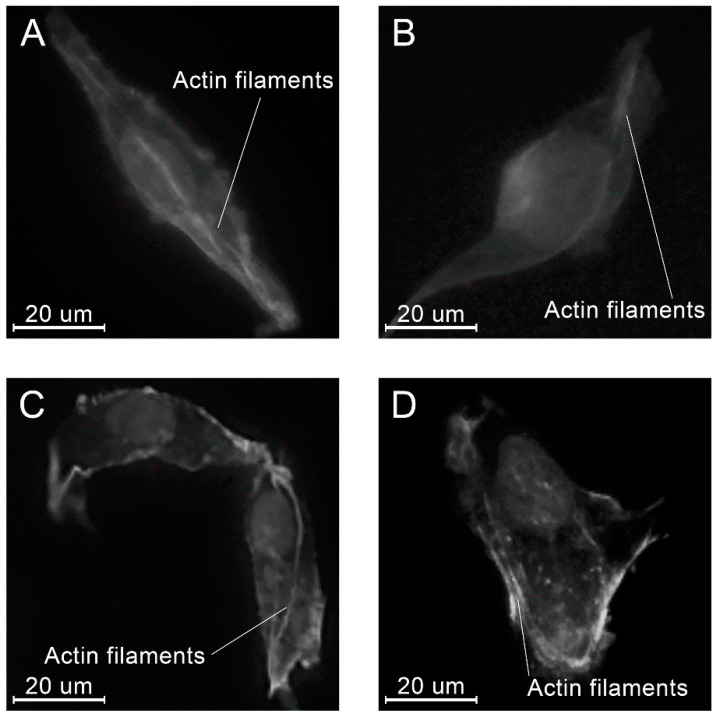
Examples of the morphology of the MG-63 cells, 8 h after seeding on the specimens of different materials: PLAMA-500 (**A**); PLAMA-1000 (**B**); PLDL (**C**); SS (**D**).

**Figure 17 molecules-26-01256-f017:**
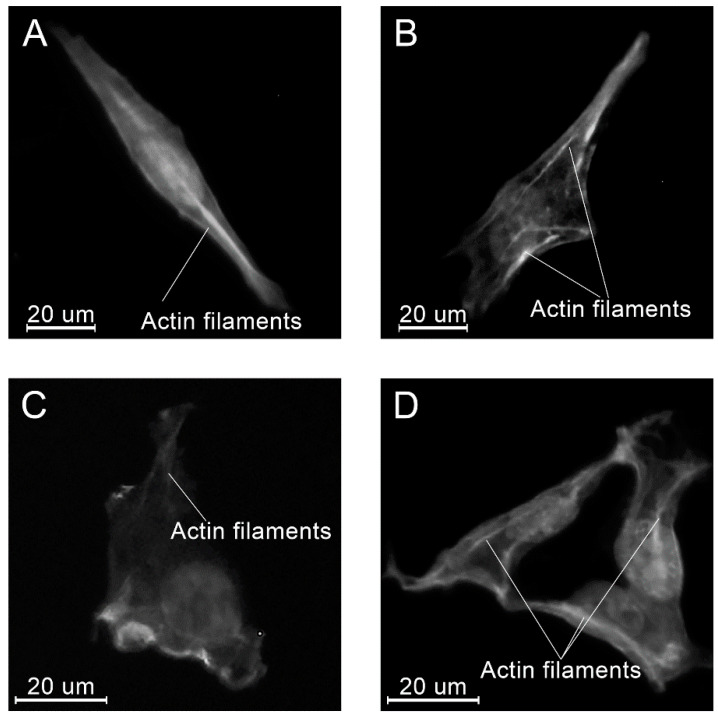
Examples of the morphology of the MG-63 cells, 24 h after seeding on the specimens of different materials: PLAMA-500 (**A**); PLAMA-1000 (**B**); PLDL (**C**); SS (**D**).

**Figure 18 molecules-26-01256-f018:**
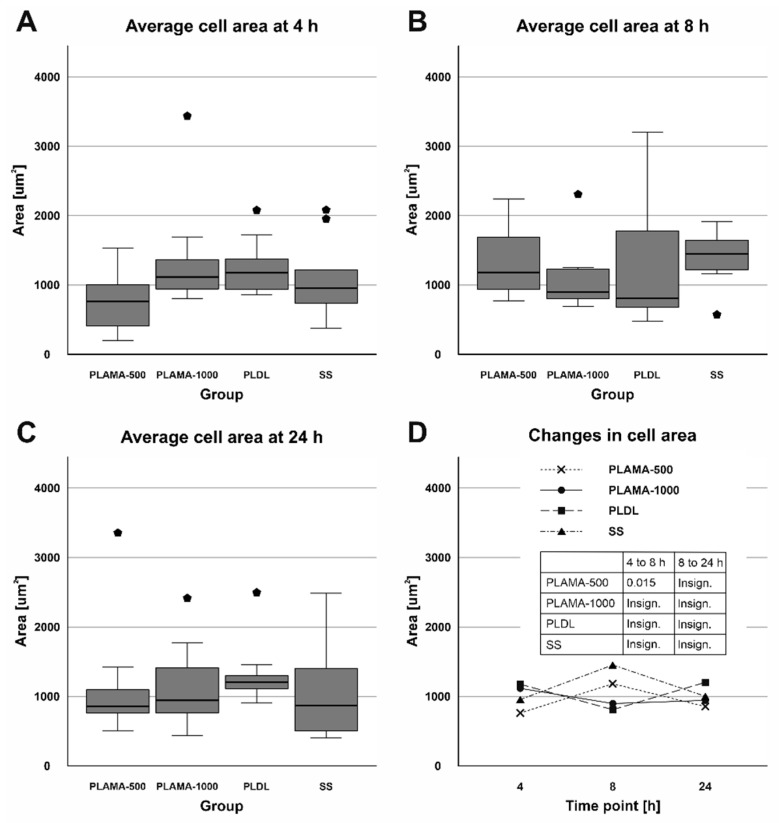
Distribution of the surface area of cells among groups at each time point (**A**–**C**); change in the surface area in all groups throughout the cell morphology test (**D**).

**Figure 19 molecules-26-01256-f019:**
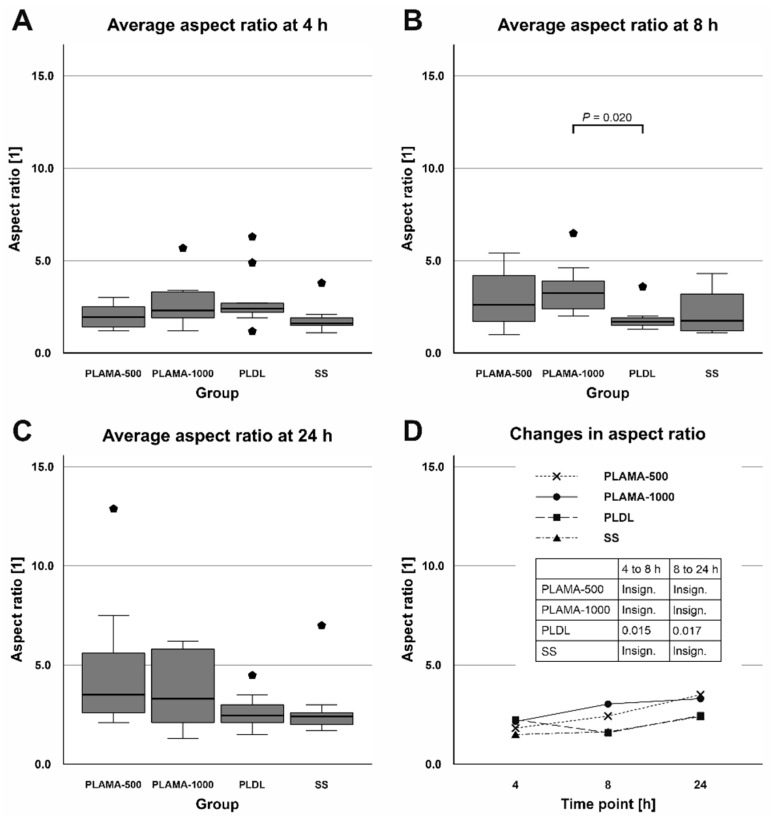
Distribution of the aspect ratio of cells among groups at each time point (**A**–**C**); change in the aspect ratio in all groups throughout the cell morphology test (**D**).

**Figure 20 molecules-26-01256-f020:**
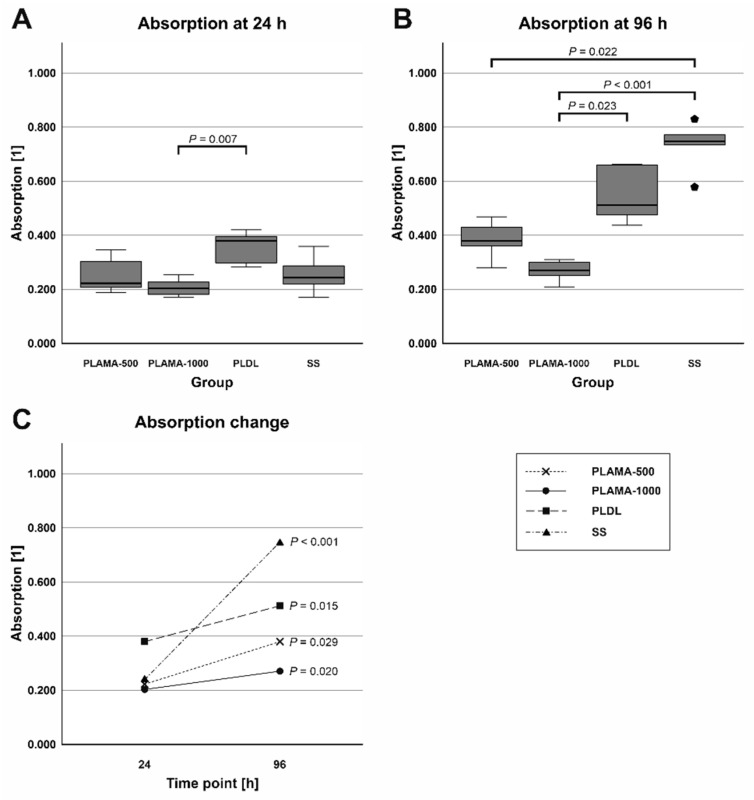
Absorbance at 450 nm measured at 24 (**A**) and 96 h (**B**) after seeding; the change in absorbance between the two time points (**C**).

**Figure 21 molecules-26-01256-f021:**
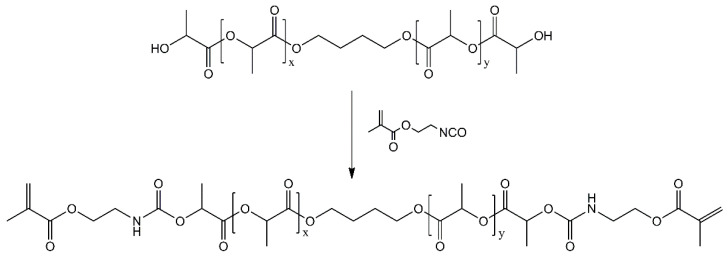
Synthesis route of PLAMA.

**Figure 22 molecules-26-01256-f022:**
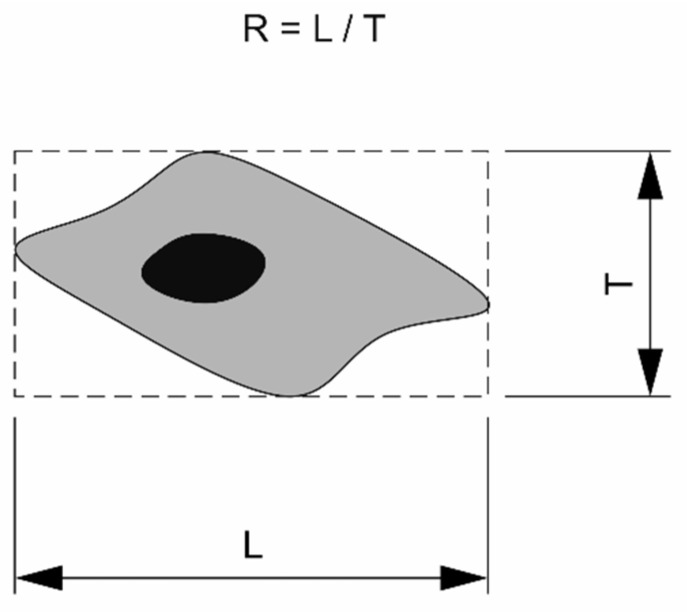
Measurement of the aspect ratio (R) of a cell.

**Table 1 molecules-26-01256-t001:** Groups of materials used in the study.

Group	Used in the Experiments:					
	Determination of Degree of Conversion	Tensile Test	Thermo-Mechanical Analysis	Biodegradation Test	Cell Morphology Test	Cell Viability Test
PLAMA-500	+	+	+	+	+	+
PLAMA-1000	+	+	+	+	+	+
PLDL ^1^	−	+	+	−	+	+
SS ^2^	−	−	−	−	+	+
BisGMA ^3^	+	+	+	−	−	−

^1^ Poly (l-/dl-) lactide (70/30%) (Corbion N.V., Netherlands). ^2^ Medical grade stainless steel 316L (Synthes AG, Switzerland). ^3^ Bisphenol-A-glycidylmethacrylate (BisGMA)/ triethylene glycol di-methacrylate (TEGDMA) co-polymer (60/40%) [57].

**Table 2 molecules-26-01256-t002:** Measured degree of conversion of double bonds in cured polymers.

Group	Degree of Conversion (%) at a Time Point (Days)				
	0	1	3	7	28
PLAMA-500	80.74 (3.10) ^1^	87.30 (1.61) ^1^	90.58 (1.51) ^1^	90.76 (0.60) ^1^	90.70 (90.20, 91.90) ^2^
PLAMA-1000	83.14 (2.42) ^1^	88.16 (2.26) ^1^	89.62 (1.38) ^1^	89.38 (0.99) ^1^	92.20 (89.75, 92.35) ^2^
bisGMA	63.00 (5.49) ^1^	79.88 (4.67) ^1^	81.20 (2.82) ^1^	82.12 (1.76) ^1^	81.50 (80.65, 84.20) ^2^

^1^ Data are given in the format: mean (SD). ^2^ Data are given in the format: median (25th percentile, 75th percentile). Five specimens were tested in each group.

**Table 3 molecules-26-01256-t003:** Numerical results of the tensile test.

Group	Number of Specimens	Elastic Modulus (MPa)	UTS (MPa)	Strain at UTS	Poisson’s Ratio
PLAMA-500	5	3350 (169)	68.5 (1.9)	0.0359 (0.0031)	0.27 (0.04)
PLAMA-1000	5	924 (68)	14.0 (1.7)	0.0344 (0.0118)	0.39 (0.03)
bisGMA	6	3995 (168)	84.2 (6.0)	0.0305 (0.0040)	0.29 (0.05)
PLDL	5	5170 (1130)	52.4 (6.9)	0.0110 (0.0027)	0.20 (0.09)

Data are given in the format: mean (SD).

**Table 4 molecules-26-01256-t004:** Numerical results of the single cycle thermomechanical analysis (TMA).

Group	Number of Specimens	Glass Transition Temperature, *T_g_* (°C)	CTE ^1^ at RT ^2^, (10^−6^/°C)	CTE at BT ^3^, (10^−6^/°C)
PLAMA-500	3	38.1 (4.3)	90.6 (9.2)	131.2 (2.0)
PLAMA-1000	3	20.8 (7.2)	146.2 (10.4)	179.9 (9.2)
bisGMA	3	109.8 (19.0)	58.9 (4.2)	75.0 (2.5)
PLDL	5	52.5 (3.5)	82.9 (5.2)	106.9 (11.9)

^1^ Coefficient of thermal expansion. ^2^ Room temperature, 18 °C–22 °C. ^3^ Body temperature, 35 °C–45 °C. Data are given in the format: mean (SD).

**Table 5 molecules-26-01256-t005:** Numerical results of the multicyclic thermomechanical analysis (TMA).

Group	Run	Glass Transition Temperature, *T_g_* (°C)		CTE at RT ^1^, (10^−6^/°C)		CTE at BT ^2^, (10^−6^/°C)	
PLAMA-500	1	44.5	42.7 (1.6) **^3^**	101.7	121.5 (20.1) **^3^**	131.3	155.9 (23.2) **^3^**
	2	42.5		121.0		159.2	
	3	41.2		141.8		177.3	
PLAMA-1000	1	12.7	11.8 (0.8) **^3^**	144.6	152.2 (14.7) ^3^	161.0	179.6 (16.3) **^3^**
	2	11.3		136.7		191.7	
	3	11.3		157.3		186.0	
bisGMA	1	124.2	113.6 (9.2) **^3^**	66.4	62.5 (3.5) ^3^	76.5	72.6 (3.4) **^3^**
	2	109.4		59.8		70.0	
	3	107.2		61.2		71.3	
PLDL	1	56.4	56.7 (0.5) **^3^**	78.4	−36.4 (5.2) ^3^	82.1	125.8 (40.6) **^3^**
	2	56.6		−277.4		162.3	
	3	57.3		89.8		133.1	

^1^ Room temperature, 18 °C–22 °C. ^2^ Body temperature, 35 °C–45 °C. ^3^ Averaged data for three runs, given in the format: Mean (SD).

**Table 6 molecules-26-01256-t006:** Mass changes in specimens after the SBF immersion test (84 days).

Group	Initial Mass [g]	Dry Mass after the Immersion [g]	Average Mass Change [g]
PLAMA-500	0.133 (0.003) ^1^	0.129 (0.003) ^1^	−0.004 (−0.004, −0.003) ^2^
PLAMA-1000	0.132 (0.002) ^1^	0.119 (0.002) ^1^	−0.013 (−0.014, −0.012) ^2^

^1^ Data are given in the format: mean (SD). ^2^ Data are given in the format: median (25th percentile, 75th percentile). Eight specimens were tested in each group.

**Table 7 molecules-26-01256-t007:** Changes in the pH value of SBF after the SBF immersion test (84 days).

Group	Number of Specimens	Average pH Value in Fresh SBF	Average pH Value after the Test
PLAMA-500	7	7.51 ^1^	7.52 (0.05) ^2^
PLAMA-1000	8		7.20 (0.02)

^1^ Measured in bulk. ^2^ Data are given in the format: mean (SD).

**Table 8 molecules-26-01256-t008:** Changes in the flexural properties of materials after the SBF immersion test.

Group		Flexural Modulus (MPa)	Ultimate Flexural Strength (MPa)	Strain at Ultimate Flexural Strength	Stress at Break (MPa)	Strain at Break
PLAMA−500	Before immersion	2458 (60) ^1^	84.4 (1.6) ^1^	0.0576 (0.0052) ^1^	83.5 (82.3, 84.0) ^2^	0.0696 (0.0188) ^1^
	After immersion	2772 (101) ^1^	94.2 (6.1) ^1^	0.0476 (0.0085) ^1^	95.0 (90.0, 99.0) ^2^	0.0476 (0.0085) ^1^
PLAMA−1000	Before immersion	333 (59) ^1^	18.0 (2.0) ^1^	0.0986 (0.0053) ^1^	N/A	N/A
	After immersion	540 (60) ^1^	20.2 (2.0) ^1^	0.0610 (0.0111) ^1^	20.0 (2.0) ^1^	0.0625 (0.0130) ^1^

^1^ Data are given in the format: mean (SD). ^2^ Data are given in the format: median (25th percentile, 75th percentile). Eight specimens were tested in each group.

**Table 9 molecules-26-01256-t009:** Numerical results of the cell morphology test: surface area of cells.

Group	Surface Area [µm^2^] at a Time Point [h]		
	4	8	24
PLAMA-500	762 (385, 1076)	1183 (911, 1781)	855 (740, 1177)
PLAMA-1000	1116 (934, 1446)	898 (789, 1238)	944 (722, 1500)
PLDL	1176 (934, 1462)	810 (667, 1828)	1203 (1104, 1338)
SS	957 (724, 1406)	1452 (1207, 1709)	872 (500, 1427)

Data are given in the format: median (25th percentile, 75th percentile). Ten specimens were tested in each group.

**Table 10 molecules-26-01256-t010:** Numerical results of the cell morphology test: aspect ratio of cells.

Group	Aspect Ratio at a Time Point [h]		
	4	8	24
PLAMA-500	2.0 (1.4, 2.5)	2.6 (1.7, 4.2)	3.5 (2.5, 6.1)
PLAMA-1000	2.3 (1.9, 3.3)	3.3 (2.4, 4.1)	3.3 (2.0, 5.9)
PLDL	2.4 (2.1, 3.3)	1.7 (1.5, 1.9)	2.5 (2.0, 3.1)
SS	1.6 (1.5, 2.0)	1.8 (1.2, 3.4)	2.4 (2.0, 2.7)

Data are given in the format: median (25th percentile, 75th percentile). Ten specimens were tested in each group.

**Table 11 molecules-26-01256-t011:** Numerical results of the cell viability test.

Group	Absorbance at a Time Point (h)	
	24	96
PLAMA-500	0.223 (0.202, 0.314)	0.380 (0.341, 0.439)
PLAMA-1000	0.203 (0.179, 0.234)	0.271 (0.241, 0.303)
PLDL	0.380 (0.294, 0.402)	0.512 (0.466, 0.660)
SS	0.243 (0.208, 0.305)	0.748 (0.695, 0.786)

Data are given in the format: median (25th percentile, 75th percentile). Six specimens were tested in each group.

**Table 12 molecules-26-01256-t012:** Spectra peaks used to control the synthesis of PLAMA oligomer mixtures.

Group	Spectra Peaks	
	IR (Neat) Wavenumbers ν (cm^−1^)	1H-NMR (CDCl3, 400 MHz) Chemical Shifts δ (ppm)
PLAMA-500	3383, 2991, 2961, 2945, 2899, 1722, 1636, 1538, 1455, 1256, 1195	6.13, 5.60, 6.06–6.29, 4.37–4.40, 4.18–4.23, 3.60–3.63, 1.96, 1.73, 1.41–1.60, 1.25, 0.88
PLAMA-1000	3415, 2992, 2959, 2945, 2901, 1754, 1636, 1534, 1453,1260, 1188	6.12, 5.60, 6.06-6.23, 4.36–4.39, 4.17–4.23, 3.49-3.61, 1.96, 1.72, 1.41-1.60, 1.26, 0.87–0.88

## Data Availability

The data presented in this study are available on request from the corresponding author. The data are not publicly available due to IPR considerations.

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
