# Peer review of "A Polymer for Application as a Matrix Phase in a Concept of In Situ Curable Bioresorbable Bioactive Load-Bearing Continuous Fiber Reinforced Composite Fracture Fixation Plates"

_molecules, 2021, doi:10.3390/molecules26051256_

Round 1
Reviewer 1 Report
Current study describes a novel material for fracture management according to the "load sharing" principle.
Regarding the clinical applications, the authors should briefly highlight the advantages of this material. For example high velocity trauma in children or economical aspects of plate removal.
Why have the authors used commercially purchased osteoblast-like cells? The proliferation properties of cells and cell lines show differences.
The viability of the cells were examined by WST analysis. However, the proliferation rate is also an important aspect in cytocompatibility studies. BrdU analysis would improve the scientific value of the paper.
Please avoid the use of personal pronouns such as we, our, etc in the text.
It is well known that the aliphatic polyesters were also used for the reconstruction of the maxillofacial defects with "load sharing" principles. (Orbita fractures etc.) Why have the authors solely focused on "load bearing"?
The fixation plates placed for load sharing are mostly used for the management of fractures with multiple segments and are usually not removed, thus the fracture healing requires a period of 6-12 Months. Therefore, the degradation properties of the material plays also a key role in clinical outcomes. It is also well known that, some biodegradable biomaterials used as fracture plates could show some lacunea formation at the fracture site, which is not optimal for fracture healing. The authors have to mention the biodegradation properties and their effects on clinical applications before suggesting the suitability of the material.
Moreover, composite fractures are often associated with involvement of soft tissues. Therefore, secondary infections is an other point to discuss. There is a lack of data regarding the anti-fouling properties on this surface. Further experiments regarding bacterial adhesion/colonisation is necessary.
Best regards.
Reviewer 2 Report
The authors have done a very thorough work on the characterization of the novel PLAMA material.
The concept of FRC fixation plate is also very interesting.
I am looking forward to see the next paper in your series.
Reviewer 3 Report
In this study, the authors proposed a concept of the in situ light curable bioresorbable bioactive load-bearing fiber reinforced composite (FRC) plate for musculoskeletal reconstruction. Briefly, a novel matrix phase polymer (named PLAMA) was synthesized. The authors focused on the investigation of the basic chemical and physical properties, as well as the biological safety with discussion on intended application. However, before the potential acceptance, some revisions are necessary as noted below.
- There are some contents that can be modified for better representation, examples are listed below.
At the begin of ‘2 Result’ part, Line 84 to Line 111, the proposed plate concept can be briefly introduced in the ‘1 Introduction’ part;
At the line 148 and line 251, the authors should avoid excessive use of descriptive language for visual characterization. If possible, the visual result can be presented by picture;
At line 159, it would be better to show the calculated equation of DC on here;
At line 281, the ‘Group PLAMA-500’ of table 7 just had 7 specimens, but authors mentioned ‘eight specimens were tested in each group’;
At line 336 to line337, the test description was not consistent with the result on table 9 and figure 18.
- Several important polymers for bio-application like peptides/haluronic acid may be considered to attract broad readerships. Recent advances on peptides as polymers for biomedical applications (VIEW 2020, 1, 20200050; VIEW 2020, 1, 20200020) are missing. Also, comparison with other common polymers, e.g. haluronic acid (Small 2019, 34, 1902441; Materials Today Bio 2019, 4, 100033), should be made to give more context. More related literatures and discussion should be added.
- The figures can be further modified and some suggestions were listed for reference:
In figure 1, bone, reinforcement phase, shell and uncured matrix phase can be showed in different colors that is more suitable for illustration. And the light-curable process can be presented. The picture of real products may be added in this figure.
In figure 2, the stage 2 in summary of physical characterization may be more suitable.
In figure 4C, the legends can be in the plots;
At line 692, the title of figure 22 seems to be abnormal;
For Table 1, the array of “type of material’ is too wide, the information can be attached to the table as supplementally instruction or explained detailly in text part, so that the table will have enough width to accommodate the experiment types. The experiment name can be listed here in abbreviation;
In Table 7 the data should be given in same format.
- Fluid resistance of this matrix phase precuring seems to be an important physical character of clinical applications, related data may be provided.
- Following the previous comments, the authors should add a few line and references about emerging application using polymers like PVP (Angew. Chem. Int. Ed., 2020, 59, 1703; Angew. Chem. Int. Ed., 2020, 59, 10831) in metabolic detection, as one of the perspectives for this work.
- There are some mistakes in this manuscript including but not limited to below, which should be corrected.
In the abstract (Line 18 of manuscript, form the same below) PLA is the abbreviation of ‘polylactide’ that firstly mentioned in the paper, so it would be better to provide the full time in this case;
At the line 40, ‘have a good biocompatibility’ the ‘a’ could be deleted, as well as the line 725 ‘a potential’;
At the line 207 ‘Table3’, a blank character was missing;
At line 423 ‘(500 and 1000)’ the unit should be added;
At line 721 authors stated this work is the ‘first step’. It can be altered as ‘first stage’.
Round 2
Reviewer 1 Report
The issues emphasized were adressed adequately.
Thenk you for giving me the opportunity to review this work.
Reviewer 3 Report
The authors addressed only parts of comments. Notably, the following two major aspects should be improved before potential acceptance.
- There are too many descriptive words without specific value or ranges. For one example, the authors used “some” many times, most of which should be clearly defined. For another example, the authors used “more” many times, most of which should be demonstrated with specific values. Furthermore, statical analysis should be provided, regarding all comparisons, to validate the technical soundness of this work.
- The references should be largely improved. Over 80 papers were cited, whereas only ~5 were in recent years (2019-now). Old literatures can be removed, while more current progress and highlights on polymers should be included and not limited to peptides/HA/PVP as the previous suggestions (VIEW 2020, 1, 20200050; VIEW 2020, 1, 20200020; Small 2019, 34, 1902441; Materials Today Bio 2019, 4, 100033; Angew. Chem. Int. Ed., 2020, 59, 1703; Angew. Chem. Int. Ed., 2020, 59, 10831).
